# Secreted Protein Acidic and Rich in Cysteine (*SPARC)* Polymorphisms in Response to Neoadjuvant Chemotherapy in HER2-Negative Breast Cancer Patients

**DOI:** 10.3390/biomedicines11123231

**Published:** 2023-12-06

**Authors:** Cristina Arqueros, Juliana Salazar, Alberto Gallardo, Marta Andrés, Ariadna Tibau, Olga Lidia Bell, Alícia Artigas, Adriana Lasa, Teresa Ramón y Cajal, Enrique Lerma, Agustí Barnadas

**Affiliations:** 1Department of Medical Oncology, Hospital de la Santa Creu i Sant Pau, 08041 Barcelona, Spain; carqueros@santpau.cat (C.A.); mandresg@cst.cat (M.A.);; 2Department of Medicine, Faculty of Medicine, Universitat Autònoma de Barcelona, 08035 Barcelona, Spain; 3Translational Medical Oncology Laboratory, Institut d’Investigació Biomèdica Sant Pau (IIB-Sant Pau), Institut de Recerca Sant Pau—CERCA Center, 08041 Barcelona, Spain; 4Institut d’Investigació Biomèdica Sant Pau (IIB-Sant Pau), Institut de Recerca Sant Pau—CERCA Center, 08041 Barcelona, Spain; agallardoa@santpau.cat (A.G.);; 5Department of Pathology, Hospital de la Santa Creu i Sant Pau, 08041 Barcelona, Spain; 6Department of Morphological Sciences, Faculty of Medicine Universitat Autònoma de Barcelona, 08035 Barcelona, Spain; 7Genetics Department, Hospital de la Santa Creu i Sant Pau, 08041 Barcelona, Spainalasa@santpau.cat (A.L.); 8Centro de Investigación Biomédica en Red en Enfermedades Raras (CIBERER), Instituto de Salud Carlos III, 28029 Madrid, Spain; 9Centro de Investigación Biomédica en Cáncer (CIBERONC), Instituto de Salud Carlos III, 28029 Madrid, Spain

**Keywords:** SPARC, breast cancer, polymorphisms, biomarker, neoadjuvant therapy

## Abstract

Secreted protein acidic and rich in cysteine (SPARC) expression has been proposed as a prognostic and predictive biomarker for some cancer types, but knowledge about the predictive value of *SPARC* polymorphisms in the context of neoadjuvant therapy for breast cancer (BC) is lacking. In 132 HER2-negative BC patients treated with neoadjuvant chemotherapy, we determined polymorphisms in the *SPARC* gene and analyzed their association with outcome. We also determined SPARC protein expression in tumor tissue. *SPARC* rs19789707 was significantly associated with response to treatment according to the Miller and Payne system in the breast (multivariate: odds ratio (OR), 3.81; *p* = 0.028). This association was significant in the subgroup of patients with luminal tumors (univariate: *p* = 0.047). Regarding survival, two *SPARC* variants showed significant associations with event-free survival: the rs19789707 variant in the subgroup of luminal A tumors (univariate: *p* = 0.006), and the rs4958487 variant in the subgroup of luminal B tumors (univariate: *p* = 0.022). In addition, *SPARC* rs4958487, rs10065756, and rs12153644 were significantly correlated with SPARC protein expression. Our findings suggest that *SPARC* polymorphisms could be good predictors of treatment response and survival in BC patients treated with neoadjuvant chemotherapy, especially those with luminal tumors.

## 1. Introduction

Sequential chemotherapy regimens with anthracyclines and taxanes achieve pathological complete response (pCR) rates of between 26% and 34% in the neoadjuvant treatment of breast cancer. These rates are higher than those obtained previously with the use of anthracyclines [1,2]. However, the value of pCR as a surrogate parameter for long-term survival in neoadjuvant chemotherapy remains controversial. The risk of relapse in patients with triple-negative and HER2-positive breast cancer is higher if they do not reach a pCR [3,4]. The association between pCR and long-term results is stronger in patients with more aggressive disease [5]. However, the analysis of pooled data from most clinical trials has only identified pCR as a surrogate endpoint for event-free survival (EFS) and overall survival (OS) in patients with aggressive tumors, raising concerns about its prognostic value [6]. Luminal A/B breast cancer continues to be a challenge, as the pCR rates are usually low and the prognostic value of pCR is uncertain. In addition, there is no clear consensus about whether the best treatment is hormonotherapy, chemotherapy, or monotherapy [7,8]. Better knowledge of the markers of long-term survival in neoadjuvant chemotherapy is therefore essential.

Cancer progression involves molecular interactions between malignant cells and their extracellular microenvironment. These interactions, which contribute to tumor growth and protection from inflammatory response, are mainly mediated by matricellular proteins, including secreted protein acidic and rich in cysteine (SPARC) [9]. SPARC has been shown to interact with extracellular matrix (ECM) components [10,11] and with growth factors [12,13]. SPARC also regulates matrix metalloproteinase expression and cytoskeleton architecture of certain cell types, supporting its role in the regulation of cell adhesion to ECM components [12,14].

Interestingly, high SPARC expression has been described in the stroma adjacent to certain tumorigenic cells [15,16]. Furthermore, it has been correlated with disease progression and poor prognosis [17,18], implying its participation in tumor progression. Nevertheless, the function of SPARC appears to vary among cancer types, and its role in breast cancer progression is controversial. Several studies have shown that high SPARC expression is associated with worse prognosis [19,20,21]. However, as SPARC facilitates the transport of nab-paclitaxel, its expression level has been proposed as a predictor of efficacy for chemotherapy schemes with nab-paclitaxel [22]. Nevertheless, the results of clinical trials that determined SPARC expression did not show better treatment responses in patients whose tumors expressed SPARC [23,24].

Most previous studies are based on tissue expression and have the limitation that available samples do not always fully represent the stromal tissue. In this setting, genomic DNA is a reliable and readily available alternative material that provides reproducible results. To our knowledge, polymorphisms in the *SPARC* gene have not yet been fully explored in the context of neoadjuvant chemotherapy in breast cancer. Here we aimed to evaluate whether variants in the *SPARC* gene predict clinical outcomes in HER2-negative breast cancer patients treated with neoadjuvant chemotherapy and to determine the association with SPARC protein expression.

## 2. Materials and Methods

### 2.1. Study Population

The study included 132 female Caucasian patients with HER2-negative infiltrating breast carcinoma treated with neoadjuvant chemotherapy including anthracycline and taxane between 2011 and 2017 at Hospital de la Santa Creu i Sant Pau (HSCSP). The patients received epirubicin 90 mg/m^2^ plus cyclophosphamide 600 mg/m^2^ every 21 days for a total of 4 cycles, followed by paclitaxel 80 mg/m^2^ for 12 weeks. We excluded patients with HER2 overexpression and those treated with neoadjuvant hormone therapy.

### 2.2. Study Design and Outcome Evaluation

In this pharmacogenetic and retrospective case-control association study, frequencies of the *SPARC* variants in the cases (patients who had had a poor outcome to therapy) were compared with those in the controls (patients who had had a good outcome to the therapy). The primary endpoint for the study was pCR, and secondary endpoints were EFS and OS [25]. After the patients completed neoadjuvant chemotherapy, pCR was defined as no residual invasive cancer in the breast and axillary nodes with presence or absence of in situ cancer (ypT0/is ypN0 or ypT0 ypN0) after surgery [26]. The tumor response was evaluated in the surgical specimen using the Miller and Payne system [27] and the residual burden cancer (RCB) [28] and classified into several categories (RCB: 0 vs. I/II vs. III; Miller and Payne: in the breast 1/2/3 vs. 4/5, and in the axilla A/D vs. B/C). Tumors with pCR, Miller and Payne 4/5 and A/D, and RCB 0 were classified as responders, while non-pCR, Miller and Payne 1/2/3 and B/C, and RCB I/II and III were classified as non-responders. For survival evaluation, EFS was defined as the time from the onset of neoadjuvant treatment until local or contralateral relapse, distant progression, death by any cause, or last clinical follow-up, whichever occurred first. OS was defined as the time from diagnosis to last clinical follow-up or death from any cause.

We used the 2011 St Gallen Consensus [29] and data from Cheang MCU et al. [30] to classify molecular subtypes. Surrogate markers for luminal A subtype tumors were positive estrogen receptors (ER) and/or progesterone receptors (PR), HER2-negative, and a low ki-67 index at a cut-off point of <14%. For luminal B subtype tumors, the surrogate markers were positive ER and/or PR, HER2-negative, and a high ki-67 index (values ≥ 14%). Triple-negative subtype tumors were those that presented absence of ER, PR expression, and HER2-negative expression/amplification.

### 2.3. Genetic Studies

We studied the 8 single-nucleotide polymorphisms (SNPs) of the *SPARC* gene analyzed in our previous study [31] (Table 1). In brief, we used the Haploview 4.2 software (v.4.2) [32] and selected the SNPs with a cut-off point of 0.8 for the r2 coefficient and minor allele frequency (MAF) over 0.05 in the European population according to the 1000 Genomes Project [33]. Genomic DNA was extracted from whole blood using the Autopure kit (Qiagen, Hilden, Germany). SNPs were genotyped by allelic discrimination on the BioMark^TM^ equipment, using SNP-type assays designed for the *SPARC* gene and Fluidigm 48.48 dynamic chips (Fluidigm, San Francisco, CA, USA). All samples and tests were prepared according to the manufacturer’s instructions. The allele frequencies of the genetic variants were comparable to those reported in the 1000 Genomes Project for the European population. All the genotypic frequencies for each SNP were in Hardy–Weinberg equilibrium except for rs967527, which was therefore eliminated from the analyses. We used the Regulome database (DB; v2.0.3) [34] and Haploreg (v4.1) [35] to infer the effect of the non-coding variants. Linkage disequilibrium (LD) analyses were performed using data from the 1000 Genomes Project.

### 2.4. Immunohistochemical Studies

SPARC protein expression was determined on the 45 (34%) available archival core needle biopsy samples collected prior to neoadjuvant chemotherapy treatment. Serial 5 µm formalin-fixed paraffin-embedded sections were cut and stained using the Envision method (Dako, Glostrup, Denmark). We used the mouse anti-human SPARC monoclonal antibody ON1-1 (Invitrogen, Vacaville, CA, USA). Immunohistochemical stains were examined independently by two of the authors (AG and CA) and discordant results were reviewed for mutual consensus. SPARC immunostaining scores in the stroma and in the epithelium were calculated by multiplying the percentage of labeled cells by the intensity of the staining (H-score; range 0–300) [36].

### 2.5. Statistical Analyses

Binary variables were analyzed using binary logistic regression, and clinical response was determined using ordinal logistic regression. The results were expressed as odds ratios (OR) and 95% confidence intervals (CIs). Survival and time to recurrence were represented with Kaplan–Meier curves and expressed as percentages. Differences in OS and EFS were analyzed using the long-rank test and Cox proportional hazards regression, and the results were expressed as hazard ratios (HRs) and 95% CIs. Histologic grade (G3 vs. G1/G2), tumor size (>2 cm vs. ≤2 cm), lymph node status (N+ vs. N0), and molecular subtype (luminal A, luminal B, triple-negative) were included as covariates in the multivariable analysis. Hardy–Weinberg equilibrium was evaluated using the Chi-square test. Codominant, dominant, and recessive models of inheritance were considered. The sample size had a statistical power of 80% to detect genetic effect sizes of moderate magnitude (OR ≤ 3) for treatment response with a two-sided 95% CI (Epi Info 7^TM^). Statistically significant associations were considered when *p*-values were <0.05. Statistical analyses were performed using STATA 15.1 statistical package, R program (v.3.2.3), SPSS statistical software (v26.0, IBM, New York, NY, USA), and haplotype analyses using the statistical package PLINK (v1.07.2) [37]. Patient selection and data analysis were carried out following the reporting recommendations for tumor marker prognostic studies (REMARK) guidelines [38].

## 3. Results

### 3.1. Clinical Results

Table 2 shows the baseline clinical and pathological data of the patients included in the study. Their median age at diagnosis was 54.4 years. Of the patients, 25% were classified as triple-negative, 27% as luminal A, and 48% as luminal B. After neoadjuvant treatment, 16% of the patients reached a pCR: 48% were triple-negative, 9% were luminal A, and 43% were luminal B. According to the Miller and Payne grading system, 11% of the lesions were grade 1, 15% were grade 2, 43% were grade 3, 14% were grade 4, and 17% were grade 5. According to the RCB class, 16% were class 0, 10% were class I, 50% were class II, and 24% were III.

The median follow-up was 62 (10–115) months. Eleven (8%) patients had a local recurrence or distant progression and six (5%) patients died.

pCR rates were highest in patients with grade 3 tumors (*p* = 0.038), in hormone receptor-negative patients (*p* = 0.009), in patients with ki-67 indexes ≥ 14% (*p* = 0.017), and in patients with negative lymphovascular invasion (*p* = 0.016).

### 3.2. SPARC Polymorphisms and Clinical–Pathological Characteristics

We identified associations between *SPARC* polymorphisms and clinical–pathological characteristics (Appendix A). For the rs12153644 variant, the A allele was associated with a higher probability of hormone receptor-positive status; 83% of patients with the AA genotype and 80% with the TA genotype were hormone receptor-positive compared to 63% of patients who were homozygous for the T allele (TT vs. TA *p* = 0.046; TT vs. AA *p* = 0.124). For the rs4958487 variant, 81% of patients with the GG genotype had a positive lymph node status compared to 65% of the heterozygous patients and 49% of the patients with the AA genotype (AA vs. AG *p* = 0.097; AA vs. GG *p* = 0.017). For the rs3210714 variant, 7% of patients with the GG genotype presented lymphovascular invasion compared to 23% of the heterozygous patients and 27% of patients with the AA genotype (GG vs. GA *p* = 0.043; GG vs. AA *p* = 0.031). A similar result was observed with rs19789707; 29% of the patients with the AA genotype presented lymphovascular invasion compared to 16% of the heterozygous patients and 8% of the patients with the GG genotype (AA vs. AG *p* = 0.124; AA vs. GG *p* = 0.043).

### 3.3. SPARC Polymorphisms and Pathological Response

Analysis of *SPARC* polymorphisms regarding the pathological response showed a correlation between the response assessed using the Miller and Payne scale in the breast (1/2/3 vs. 4/5) and the rs19789707 variant (*p* = 0.07 in a recessive model). This correlation reached statistical significance when adjusting for clinical covariates (OR, 3.81; 95% CI, 1.15–12.56; *p* = 0.028). A total of 84% of patients with the GG genotype did not achieve a pathological response compared to 65% of patients carrying the A allele. When the molecular subtype was considered, we found a statistically significant association in the luminal subgroup (*p* = 0.047). This variant did not present any association with respect to the axillary response either in the total group or in the subgroup of luminal tumors. When analyzing the relationship between the rs19789707 variant and the pathological response according to RCB, we observed a lower response in the heterozygous patients (multivariate: AA vs. AG OR, 2.22; 95% CI, 5.00–1.01; *p* = 0.048; and AA vs. GG OR, 1.21; 95% CI, 3.23–0.45; *p* = 0.704).

Additionally, in the univariate analysis, we found that patients with the GG genotype for the rs4958487 variant had a lower probability of achieving a pathological response assessed by Miller and Payne in the axilla than patients with the AG or AA genotypes (GG 67% vs. AG 51% vs. AA 41%; AA vs. AG *p* = 0.293; AA vs. GG *p* = 0.064), although this finding was not statistically significant.

Haplotype analyses showed significant associations between the rare haplotype GACTCTG (rs3210714| rs1978707| rs2347128| rs17718347| rs10065756| rs12153644| rs4958487) (frequency = 2%) and pCR (*p* = 0.02). We also observed a significant association between this haplotype and local response as determined by the Miller and Payne grading system in the breast (*p* = 0.02). Our results were similar considering the luminal subgroup (*p* = 0.007 and *p* = 0.005, respectively). The haplotype was more frequent in responders.

### 3.4. SPARC Polymorphisms and Survival

*SPARC* rs19789707 and rs4958487 variants showed associations with EFS. For the rs19789707 variant, we observed a trend towards significance; the 5-year EFS was 84% (95% CI, 57.1–94.8) in patients with the GG genotype and 94% (95% CI, 87.4–97.3%) in patients carrying the A allele (*p* = 0.089 in a recessive model). This association did not reach significance after adjusting for the covariates (HR, 2.44; 95% CI, 0.66–8.93; *p* = 0.179). However, it did show statistical significance when analyzing the luminal A subgroup (univariate: *p* = 0.022 in a codominant model; *p* = 0.006 in a recessive model) (Figure 1A). For the rs4958487 variant, we obtained no significant associations with EFS for the total cohort (*p* = 0.479), but analysis according to molecular subtype showed a statistical significance for the luminal B subtype. Patients with the GG genotype had a shorter EFS than the other patients (univariate: *p* = 0.066 in a codominant model; *p* = 0.022 in a recessive model) (Figure 1B).

None of the SNPs analyzed showed significant associations with OS. However, the rs19789707 variant showed a trend towards significance (*p* = 0.069); the 5-year OS was 100% (95% CI, not applicable (NA)) in heterozygous patients compared to 95% (95% CI, 81.9–98.8%) in the AA genotype patients and 91% (95% CI, 68.8–97.7%) in the GG genotype patients.

### 3.5. SPARC Polymorphisms and SPARC Expression

The median SPARC expression values in tumor tissue, quantified using the H-score system (0–300), were 120 in epithelium and 80 in the stroma (r = 0.530; *p* < 0.001). We observed numerical differences between luminal (80) and triple-negative subtypes in the stroma (150) (*p* = 0.254).

In the analysis between *SPARC* polymorphisms and SPARC expression, we identified correlations both in the stroma and the epithelium for rs10065756 (stroma: *p* = 0.046 and epithelium: *p* = 0.068), rs121553644 (stroma: *p* = 0.010 and epithelium: *p* = 0.007), and rs4958487 (stroma: *p* = 0.005 and epithelium: *p* = 0.005) variants (Table 3). The rs17718347 variant was correlated only in the epithelium (*p* = 0.030).

### 3.6. SPARC Expression and Clinical Outcomes

In the analysis of SPARC expression and clinical outcomes, we did not find any association with pathological response or with survival.

### 3.7. In Silico Analysis

In silico analysis was only performed for rs1978707 and rs4958487, as these two variants showed associations with the study outcomes. RegulomeDB assigned a score of 5 to the rs1978707 (intron 4) variant, suggesting, at most, its minimal role in regulating gene expression. Nevertheless, this variant was in LD with 15 SNPs, 2 of which had a score of 1f: rs725937 (D′ = 1; r^2^ = 0.95) and rs7719521 (D′ = 1; r^2^ = 1). The low score in these cases indicates that these SNPs may affect gene expression and may also alter the binding domain of transcription factors, rs725937 (E2F, Irf, SIX5), and rs7719521 (Spz1). Regarding the rs4958487 (intron 1) variant, it was assigned a score of 4, so although it may alter possible transcription factor binding domains (Duxl, Pbx3), its functionality was not evident.

## 4. Discussion

SPARC expression has been extensively analyzed as a prognostic and treatment response biomarker in several cancer types, including breast cancer. However, as far as we know, this is the first pharmacogenomic study to assess the predictive value of *SPARC* polymorphisms in a neoadjuvant chemotherapy setting. We found that *SPARC* rs1978707 and rs4958487 variants were associated with treatment response and survival in HER2-negative breast cancer patients, especially in the luminal subtype.

SPARC is a matricellular protein that participates in the activation of the epithelium–mesenchyme transition through the AKT pathway in some types of cancer [39]. It also participates in the immune response and in malignant transformation processes [40,41,42,43]. Variants in the *SPARC* gene have been identified as susceptibility factors [44,45] and as predictor and prognostic biomarkers in some cancers [31,46]. *SPARC* variants as biomarkers for breast cancer risk and prognosis have only been previously proposed in a case–control study [47]. Using the Nottingham prognostic index (NPI), they classified patients as having either moderate or poor prognosis and found that *SPARC* rs7719521 was associated with the NPI and VEGF expression.

We analyzed polymorphisms in the *SPARC* gene and found that *SPARC* rs1978707 was associated with a lower probability of achieving a pathological response. We also observed this association for the luminal subtype. Interestingly, we identified a rare haplotype associated with a pathological response in the overall population and in the luminal subgroup, but validation in a larger sample is needed. We found significant associations between the rs1978707 and rs4958487 variants and a higher risk of relapse in the luminal subgroup. *SPARC* rs1978707 is an intron variant that is in LD with two other SNPs that could modify SPARC expression, rs725937 and rs7719521. The latter was the variant identified in the study of Bawazeer et al. [47], suggesting that it may have a role in breast cancer. In our study, *SPARC* rs4958487 was associated with tumor protein expression; patients with the GG genotype presented lower protein expression, both in the stroma and in the epithelium. We also found correlations with three other *SPARC* variants and SPARC expression, suggesting that the variants could provide information regarding protein expression in tumor tissue. The potential and possible mechanisms through which the *SPARC* gene may play a role in BC are currently unknown. We hypothesize that in luminal tumors, genetic variants may alter the function of certain proteins expressed intracellularly or in the tumor microenvironment, such as SPARC, and, consequently, may prevent the effects of standard chemotherapy treatment on the tumor cells, leading to a worse prognosis [48,49,50,51,52].

It has not yet been confirmed whether pCR is a prognostic surrogate for patients receiving neoadjuvant chemotherapy for luminal subtype breast cancers. Von Minckwitz G et al. [5] suggested that pCR is a potent surrogate marker of prognosis in most patients with breast cancer but not for ER-positive tumors. They also demonstrated that pCR was predictive of good survival in ER-positive tumors with high tumor proliferation [53]. Our finding that *SPARC* variants could predict response and survival in the luminal subtype suggests these SNPs could facilitate treatment selection in these patients.

Currently, data regarding the value of SPARC expression as an outcome biomarker for the various molecular subtypes of breast cancer is limited and inconsistent [22,23,51,54]. While some studies have observed that high SPARC expression was associated with poor prognosis and worse EFS and OS in several histological types [19,20,21,50,55], other studies have shown an association between low levels of SPARC and worse survival [56,57]. In their recent meta-analysis considering SPARC expression and prognosis in breast cancer, Shi et al. [57] showed that low SPARC expression correlated with worse overall and distant metastasis survival rates in grade 1/2 tumors, HER2-positive tumors, and luminal A subtype tumors. However, the EFS was better in the luminal B subgroup. The discrepancies observed between these studies are likely due to a lack of direct correlation between SPARC protein expression and its mRNA levels, since they depend on transcriptional and translational regulation processes and on mRNA and protein degradation. Disappointingly, in our explorative analyses, we did not find differences in SPARC protein expression with respect to treatment response or survival in luminal and triple-negative tumors, probably due to the limited number of samples available for the immunohistological study.

SPARC expression has been described in the stroma adjacent to the tumor epithelium, revealing its possible involvement in breast cancer invasion [58]. Nonetheless, most previously mentioned studies did not determine SPARC expression in stromal cells, which may have limited its predictive value. Results from studies conducted in SPARC-null mice suggest that SPARC expression in the surrounding tissues may regulate tumor growth [59,60]. In the present study, we identified a correlation between SPARC expression in the epithelium and in the stroma, providing evidence that SPARC may be important in tumor–host interactions between breast cancer cells and stromal fibroblasts. This observation is in agreement with several studies showing that tumor cells probably mediate a paracrine effect that induces the expression of SPARC by means of neighboring stroma, a process in which exosomes could intervene [61].

Our exploratory study has several limitations. First, the small sample size, the additional stratification of the analyses according to the molecular subtype, and the retrospective design probably influenced the strength of the results. However, the study provides additional evidence to support the importance of determining genomic variants as predictors of outcome in breast cancer. Second, the limited availability of tumor tissue for research purposes clearly influenced the ability to reproduce the SPARC expression associations reported in previous studies. Notwithstanding, our findings allowed us to describe correlations between *SPARC* polymorphisms and their expression in stromal and epithelial cells. Third, we note the lack of a control group to discern whether the pCR rate and the higher rate of response were due to the chemotherapy regimens used.

## 5. Conclusions

Our study shows that *SPARC* polymorphisms may have a prognostic and predictive value in breast cancer. Pre-therapeutic analysis of SPARC in blood samples could facilitate the selection of patients for neoadjuvant therapy, especially for those with luminal breast cancer subtypes, and consequently improve long-term survival.

The future directions of our study include the two *SPARC* variants that could be good predictors of outcome in luminal breast cancer patients treated with neoadjuvant chemotherapy. The integration of these variants into a prospectively validated predictive model could help us select patients who are more likely to have a better response and survival. These data could also allow us to de-escalate or escalate treatments and select patients who could benefit from recently approved treatments such as the immune checkpoint inhibitors, PARP inhibitors, and selective CDK4/6 inhibitors.

## Figures and Tables

**Figure 1 biomedicines-11-03231-f001:**
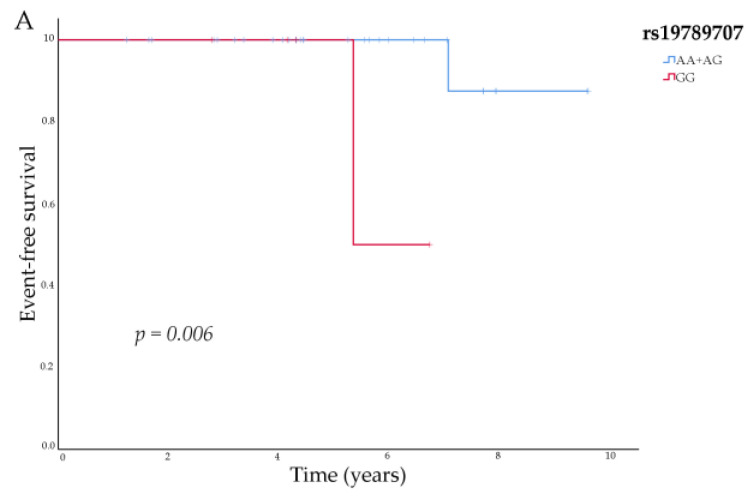
Event-free survival according to (**A**) *SPARC* rs19789707 variant in patients with luminal A breast cancer; (**B**) *SPARC* rs4958487 variant in patients with luminal B breast cancer.

**Table 1 biomedicines-11-03231-t001:** *SPARC* single-nucleotide polymorphisms.

Reference SNP	SNP Label	SNP Localization	MAF
rs4958487	c.-14+2752T>C	Intron 1	0.41
rs12153644	c.-13−4174A>T	Intron 1	0.33 ^
rs10065756	c.-13−3184G>T	Intron 1	0.33
rs17718347	c.-13−3131G>A	Intron 1	0.41
rs2347128	c.-13−1945G>C	Intron 1	0.47
rs967527	c.58−484A>G	Intron 2	0.11
rs1978707	c.208+31C>T	Intron 4	0.43
rs3210714	c.*1200G>A	Exon 10 (3’-UTR)	0.46

*SPARC*: secreted protein acidic and rich in cysteine gene; SNP: single-nucleotide polymorphism; MAF: minor allele frequency derived from the 1000 Genomes Project phase 3 sequence data for the European population, except for rs12153644; ^ MAF reported in the HapMap-CEU population (accession date: 19 July 2022). Label according to the accession number NM_003118.3 (SPARC).

**Table 2 biomedicines-11-03231-t002:** Patients’ baseline clinical and pathological characteristics (N = 132).

Characteristic	N (%)
**Age**	
<50	48 (36)
≥50	84 (64)
**Menopausal status**	
Premenopausal	53 (40)
Postmenopausal	71 (54)
Perimenopausal	8 (6)
**Tumor size**	
T1	9 (7)
T2	58 (44)
T3	37 (28)
T4	28 (21)
**Tumor type**	
Ductal	113 (86)
Lobular	12 (9)
Mixed (ductal and lobular)	3 (2)
Other histologic type	4 (3)
**Histologic grade**	
G1	8 (6)
G2	86 (65)
G3	38 (29)
**Lymphovascular invasion**	
Negative	107 (81)
Positive	25 (19)
**Perineural invasion**	
Negative	120 (91)
Positive	12 (9)
**ki-67 index**	
<14%	42 (32)
≥14%	90 (68)
**Clinical N-stage**	
cN0	49 (37)
cN+	83 (63)
**Estrogen receptor status**	
Positive	97 (74)
Negative	35 (26)
**Progesterone receptor status**	
Positive	81 (61)
Negative	51 (39)
**Pathological complete response (pCR)**	
No pCR	111 (84)
pCR	21 (16)

**Table 3 biomedicines-11-03231-t003:** Associations between *SPARC* polymorphisms and SPARC protein expression in tumor tissue.

SNPs	Epithelium		Stroma	
	SPARC Median H-Score (Range)	*p*-Value	SPARC Median H-Score (Range)	*p*-Value
rs10065756				
CC	160.0 (120.0, 195.0)	0.068	120.0 (75.0, 145.0)	**0.046**
AC	100.0 (40.0, 180.0)		40.0 (20.0, 140.0)	
AA	40.0 (20.0, 60.0)		100.0 (80.0, 120.0)	
rs12153644				
TT	180.0 (150.0, 240.0)	**0.007**	140.0 (120.0, 160.0)	**0.010**
TA	120.0 (40.0, 180.0)		60.0 (20.0. 140.0)	
AA	60.0 (40.0, 60.0)		80.0 (40.0, 80.0)	
rs17718347				
TT	160.0 (120.0, 180.0)	**0.030**	120.0 (70.0, 140.0)	0.161
TC	120.0 (40.0, 180.0)		40.0 (20.0, 140.0)	
CC	40.0 (12.5, 60.0)		100.0 (42.5, 130.0)	
rs19789707				
AA	90.0 (60.0, 160.0)	0.832	60.0 (20.0, 120.0)	0.498
AG	120.0 (60.0, 180.0)		100.0 (40.0, 140.0)	
GG	155.0 (70.0, 210.0)		100.0 (70.0, 160.0)	
AA + AG ^a^	120.0 (60.0, 180.0)	0.570	80.0 (20.0, 140.0)	0.278
rs2347128				
CC	155.0 (70.0, 180.0)	0.350	110.0 (70.0, 140.0)	0.540
CG	155.0 (50.0, 180.0)		70.0 (20.0, 145.0)	
GG	80.0 (40.0, 160.0)		60.0 (20.0, 120.0)	
rs3210714				
GG	150.0 (70.0, 180.0)	0.623	100.0 (70.0, 140.0)	0.476
GA	160.0 (40.0, 180.0)		60.0 (20.0, 140.0)	
AA	80.0 (60.0, 160.0)		80.0 (20.0, 140.0)	
rs4958487				
AA	210.0 (160.0, 240.0)	**0.005**	150.0 (120.0, 160.0)	**0.005**
AG	120.0 (40.0, 160.0)		40.0 (20.0, 140.0)	
GG	70.0 (60.0, 120.0)		80.0 (60.0, 100.0)	

SNPs: single-nucleotide polymorphisms; SPARC: secreted protein acid and rich in cysteine. ^a^ recessive model. Statistically significant *p*-values are marked in bold.

## Data Availability

The data presented in this study are available from the corresponding authors on reasonable request.

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
