# Peer review of "Secreted Protein Acidic and Rich in Cysteine (SPARC) Polymorphisms in Response to Neoadjuvant Chemotherapy in HER2-Negative Breast Cancer Patients"

_biomedicines, 2023, doi:10.3390/biomedicines11123231_

Round 1

Reviewer 1 Report

Comments and Suggestions for Authors

Reviewer 2 Report

Comments and Suggestions for Authors

Review comment

This article titled as “SPARC polymorphisms in response to neoadjuvant chemotherapy in HER2-negative breast cancer patients” has enrolled 132 HER2-negative BC patients treated with neoadjuvant chemotherapy. Authors analyzed polymorphisms in the SPARC gene and their association with outcome, determined SPARC protein expression in tumor tissue. They concluded that SPARC rs19789707 was significantly associated with response to treatment according to the Miller and Payne system in the breast (multivariate: odds ratio (OR), 3.81; P = 0.028). This association was significant in the subgroup of patients with luminal tumors (univariate: P = 0.047). Regarding survival, SPARC rs19789707 and rs4958487 variants showed significant associations with event-free survival in the subgroup of luminal tumors (univariate: P = 29 0.006 and P = 0.022, respectively). In addition, SPARC rs4958487, rs10065756 and rs12153644 were significantly correlated with SPARC protein expression. They suggested that SPARC polymorphisms could be good predictors of treatment response and survival in BC patients treated with neoadjuvant chemotherapy, especially those with luminal tumors. This study contains certain practical meaning. However, some apparent flaws still exist. Revision is recommended before acceptance.

1.“The study included 132 patients with HER2-negative infiltrating breast carcinoma treated 80 with neoadjuvant chemotherapy including anthracycline and taxane between 2011 and 2017 at 81 Hospital de la Santa Creu i Sant Pau (HSCSP). We excluded patients with HER2 overex-82 pression and those treated with neoadjuvant hormone therapy”. The specific chemo agents (including dosage and cycles) should be provided for all enrolled patients in a table. Moreover, the response rates should be recalculated based on different types of neoadjuvant chemotherapy.

2. Please comprehensively discussed the potential and possible mechanisms why SPARC polymorphisms could be good predictors of treatment response and survival in BC patients treated with neoadjuvant chemotherapy, especially those with luminal tumors. Authors should also provide related figures to display the mechanisms for readers.

3. The future perspective such as the future research of SPARC polymorphisms should be discussed in the section of DISCUSSION, based on authors’ conclusion. 

4. (DOI: 10.3736/jcim20110511 ) is recommended to be cited after “SPARC is a matricellular protein that participates in the activation of the epithelium-273 mesenchyme transition through the AKT pathway in some types of cancer”.

Reviewer 3 Report

Comments and Suggestions for Authors

1. 132 patients, female and male information need to be given.

2. All the genotypic frequencies for each SNP were in Hardy-Weinberg equilibrium, except for rs967527 which was therefore eliminated from the analyses. Why it was eliminated.

3. Please explain the a prognostic and predictive point of view,which need to corresponding to the related results.

Round 2

Reviewer 1 Report

Comments and Suggestions for Authors

The study design and description of study endpoints could be more rigorous. For example, we often use the expression that ‘EFS was defined as the time from … to disease progression, death, or discontinuation of treatment for any reason, whichever occurred first.’

Reviewer 2 Report

Comments and Suggestions for Authors

Thank the authors, I am satisfied with this revision. Acceptance is recommended.

Author Response

Thank you.